# A DECISION TREE ALGORITHM FOR MDP

## ABSTRACT

Decision trees are robust modeling tools in machine learning with human-interpretable representations. The curse of dimensionality of Markov Decision Process (MDP) makes exact solution methods computationally intractable in practice for large state-action spaces. In this paper, we show that even for problems with large state space, when the solution policy of the MDP can be represented by a tree-like structure, our proposed algorithm retrieves a tree of the solution policy of the MDP in computationally tractable time. Our algorithm uses a tree growing strategy to incrementally disaggregate the state space solving smaller MDP instances with Linear Programming. These ideas can be extended to experience based RL problems as an alternative to black-box based policies.

## 1 INTRODUCTION

Deep neural network based Reinforcement Learning (RL) has seen recent success in tackling Markov Decision Problem (MDP) instances with large state-action space. For example, the MuZero model (Schrittwieser et al. (2020)) has been able to successfully beat human performance in previously computationally untractable problems such as Shogi, Chess and Atari games with a single modeling paradigm. The success of these methods can be traced to two fundamental ideas in RL: First, the iterative nature of the $Q$-learning algorithm (see Watkins & Dayan (1992)) where the state-action value function can be updated locally without having to handle the whole state-action space at once as in exact methods; Second, the non-linear parameterization of the state-action value function and the policy function with deep neural networks, where in a sense, the state space model is "learned" and compressed without having to update and explore the whole state space. While successful methods, they are not devoid of drawbacks. Their strength in not taking into account the whole action-state space is also their weakness which manifests in the well-known worst case slow convergence of $Q$-learning (see Szepesvári et al. (1998)) as well as the relatively slow convergence of the gradient-based methods for training deep neural networks for regression.

In this paper, we aim to apply the essence of these two ideas to develop an algorithm to solve a subset of instances of MDP, that while having large enough state-action space such that exact methods are computationally intractable, their solution policy admits a representation as a decision tree that can be computed by exact methods. In summary, we aim to do an improvement over these instances with two ideas: First, the use of exact methods, such as Dynamic Programming (DP) or Linear Programming (LP), rather than iterative methods to compute the state-action value function; Second, the use of decision trees to partition the state space and codify the optimal policy rather than using deep neural networks. In Figure 1 we show the subset of problems we aim to tackle in the gray area. That is, problems with large enough state space such that exact methods are not tractable, but the action space is still not too large. This is key, as the size of the policy function representation as a decision tree depends on the structure of the state space being partitionable in regions with the same optimal action.

To illustrate these concepts we consider the cartpole balancing example, which consists of controlling a cart that can move either to right or left in a straight rail. Perpendicular to the axis of movement of the cart, there is a pole attached to a rotating axis at its center. With two actions $\mathcal{A} = \{\leftarrow, \rightarrow\}$ the goal is to have the pole attached to the cart balanced in the upright position for the longest possible amount of time. The state space of the system is a 4-dimensional vector of real numbers $\mathcal{S} = \{(x, v, \theta, w) : x \in [-\ell_1, \ell_1], v \in [-\ell_2, \ell_2], \theta \in [-\ell_3, \ell_3], w \in [-\ell_4, \ell_4]\}$ (position, horizontal velocity, angle and angular velocity). Here, the state space is of infinite size $|\mathcal{S}| = \infty$. Instead of using deep $Q$-learning, we use a decision tree based algorithm that outputs the policy in Figure 2.

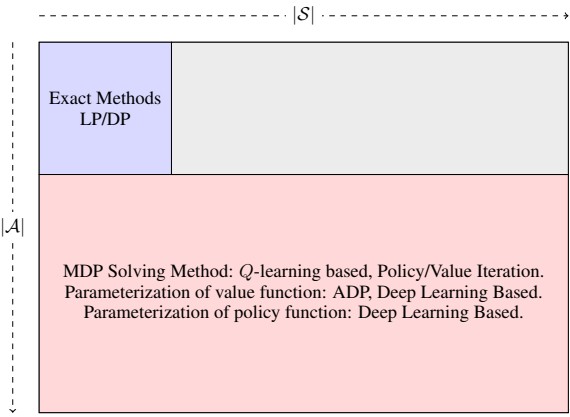

Figure 1: Solution Methods for MDP relative to state-action space size. The blue area denotes problems that are tractable using exact methods. The red area is the subset of problems where deep neural network based $Q$-learning is the most used solution method. The gray area is the subset of problems we aim to tackle.

This decision tree serves two functions simultaneously: It defines a partition or "aggregation" of the state space $\mathcal{S}$ into disjoint regions that will be the states of a much smaller MDP; and second and most importantly, it summarizes the optimal policy as a decision tree, giving a transparent and interpretable control policy. These two points are the same ideas that make deep $Q$-learning methods successful: Aggregation of the space and non-linear parameterization of the optimal policy. The MDP went from having a infinite cardinality to a reduce state space of only 6 states.

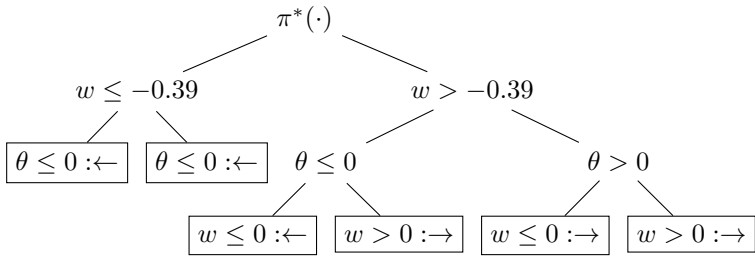

Figure 2: Cartpole solution policy $\pi^*(x, v, \theta, w)$ from our algorithm.

## 2 RELATED WORK

In the RL and MDP literature the idea of state aggregation starts as early as the discipline itself as an answer to the curse of dimensionality for large state-action spaces. One of such ideas is to make a large state space discrete by aggregating states into "boxes" (see Fox (1973)). Our method takes ideas from the Variable Resolution idea where a given discretization is further refined by constructing finer boxes around regions of the state space where the discretization does not approximate well the value function. One of those methods is the "Trie Model" (see Moore (1991)) where a tree model with similar ideas to ours is used to identify regions of the state space that need to be further partitioned while still applying Dynamic Programming methods to obtain a policy. Their main criterion to make a split was to identify regions of the state space where the model was inaccurate (in a value function sense) and increase the resolution of the MDP by further partitioning these regions. In Singh et al. (1995) the convergence for the $Q$-learning algorithm is proved for the "soft-aggregation" scheme where states belong to disjoint clusters with a given probability (including the hard cluster case where states belong to only one cluster), our work borrows a similar description of the aggregated MDP (from a cluster level) that we use to define our Linear Programming formulation. Bounds on the performance of value iteration over disjoint state aggregation are given in Van Roy (2006).

A large stream of literature then focuses on parameterization of the value function as means to defeat the curse of dimensionality (which is an indirect way to aggregate the state space). Approximate Dynamic Programming (see Bertsekas (2008) for a summary overview) is a parameterization of the value function by a lower-dimensional linear basis of the state space. On the non-linear case, nearest neighbor approximations have been used in Shah & Xie (2018). Kernel functions are used in Ormoneit & Sen (2002) and Ormoneit & Glynn (2002). Decision Trees have been used in Ernst et al. (2005) to similarly approximate the $Q$-function of unseen samples. Much of the recent literature focus on Deep Neural Network based RL which can be seen as a non-linear representation of the state-action value function (see Li (2017) for an overview) as well as a non-linear parameterization of the policy as stated in the introduction. Actor-critic architectures (see Haarnoja et al. (2018)) have also seen recent success in tackling large scale problems.

A characterization theory of state aggregations is studied in Li et al. (2006), where the most common families of aggregations (bisimulation, and equivalent value function aggregations) is studied and a hierarchy between them is given. A related work to ours is Kim & Dean (2003) where a similar sequential partition is used for finding sequential state aggregations, our work differs in the criteria to make splits, whereas we approximate the behavior of the whole MDP, Kim & Dean (2003) chooses a split based on the maximum distance between value functions from one iteration to the next positioning their work closer to the variable resolution literature.

Our method is new in the sense that instead of exclusively focusing on the resolution relative to the value function (that is, instead of trying to approximate and interpolate the value function at unseen points) we find a state aggregation abstraction of the original state space with a reduced state space with the same optimal policy as in the original state space, given that the original MDP admits such representation. We further discuss the theory and potential computational savings of MDP abstraction in Section A.3.

## 3 METHODOLOGY

In this section we describe our methodology connecting the representation of policies as decision trees and the aggregation of the original state space resulting in a much smaller MDP with equivalent optimal policy.

### 3.1 MARKOV DECISION PROCESSES

A Markov Decision Process (MDP) is a modeling framework that in simple terms finds optimal decisions with a changing environment over time with the goal of finding the "best" actions given the state of the environment. Formally, a MDP is composed of an environment represented by a state space $\mathcal{S}$, a set of actions $\mathcal{A}$. An agent in state $s \in \mathcal{S}$, takes an action $a \in \mathcal{A}$. After this, the agent transitions to state $s' \in \mathcal{S}$ and receives a reward $r \in \mathbb{R}$ with probability $p(s', r|s, a)$, which is independent of the previous history of states or actions, making the model a Markov chain. Given the state and action $s, a$ the reward is a quantity $R(s, a, s')$. A policy $\pi : \mathcal{S} \to \mathcal{A}$ is a function that maps states $s \in \mathcal{S}$ to a single action $a$[1]. For simplicity, assume that there is a terminal state that ends the subsequent transitions (an absorbing state $E$). This state is reached in a random number of $T$ transitions. The cumulative discounted reward an agent receives by interacting in the system using a policy $\pi$ is given by:

$$\mathsf{E}_\pi \left[ \sum_{t=0}^{\infty} \gamma^t R(s_t, \pi(s_t), s_{t+1}) \right]. \tag{1}$$

Given a complete description of the model, the policy that maximizes the cumulative reward is given by the Bellman optimality equation:

$$V_\pi(s) := \sum_{s',r} \mathsf{P}(s', r|s, \pi(s))[r + V_\pi(s')], \tag{2}$$

$$V_*(s) = \max_a \sum_{s',r} \mathsf{P}(s', r|s, a)[r + V_*(s')]. \tag{3}$$

---

[1]The extension to stochastic policies is straightforward and not considered here.

Where $V_\pi(s)$ is the expected reward starting at state $s$ following policy $\pi$, the expression $V_\pi(s)$ can be intuitively derived by a first-step analysis on the fact that the system is memoryless by the Markov property of Markov chains, that is, the expected reward in one state is simply the one step expected reward obtained in that state $r$ (after taking action $\pi(s)$ and transitioning to state $s'$) plus the expected reward of starting at state $s'$. The optimal policy $V_*(s)$ is simply taking the greediest action $a$ at every state $s$ in the expected reward sense.

### 3.2 Representation of Policies as Decision Trees

In this subsection we show the connection between state aggregation and policy representation as decision trees. A policy $\pi$ is a mapping between the state space $\mathcal{S}$ and the action set $\mathcal{A}$. This mapping is similar to a classification problem, in the sense that in a classification problem the goal is to divide the input space into "regions" classifying them into different classes. In MDP, the goal is similar as the policy divides the state space $\mathcal{S}$ into disjoint regions corresponding to elements of the action set $\mathcal{A}$.

Similar to Deep Neural Networks (DNN), decision trees are also good function approximators. An analogous function approximation theorem for decision trees is that for any policy $\pi(s)$, there exist a tree $\mathcal{T}$ with $J$ leaves $\mathcal{L} = \{\mathscr{E}_j\}_{j=1}^J$ such that the policy $\pi$ can be approximated by the function $\pi(s) = \sum_{j=1}^J \bar{\pi}(\mathscr{E}_j) \mathbf{1}\{s \in \mathscr{E}_j\}$ where $\{\bar{\pi}(\mathscr{E}_j)\}_{j=1}^J$ is a collection of piecewise constant actions in $\mathcal{A}$.

Given this relation between a policy and a decision tree $\pi(s) = \sum_{j=1}^J \bar{\pi}(\mathscr{E}_j) \mathbf{1}\{s \in \mathscr{E}_j\}$, it is natural to think of a related MDP where the states are the partitions of the state space rather than the larger (and potentially infinite) original state space. This approximated MDP could be far away from accurately approximate the reward of the original MDP but its resolution is good enough to identify the optimal policies.

The challenge lies in identifying a growing strategy for the trees that is able to find the optimal policy $\pi^*$ by iteratively partitioning the state space and taking the leaves of the tree as meta-states of a reduced MDP approximating the behavior of this policy in the original state space. The growing strategy needs an objective value that is able the compare the overall quality of different trees with respect to finding optimal policies in the original state space. For example, as any tree algorithm normally starts by collapsing all states into a single one, any partition into two disjoint subsets needs to take into account the sizes of the partitions as well as the approximated behavior of the aggregated MDP with respect to the original state space and quantify its improvement with respect to the initial tree where all states are collapsed into one.

### 3.3 Aggregated MDP

Suppose the state space of the original MDP is aggregated into $J$ meta-states. This aggregation is a disjoint partition $\mathcal{S} = \cup_{j=1}^J \mathscr{E}_j$ into disjoint events $\mathscr{E}_j$. Then, for any state $s \in \mathcal{S}$ there is only one index $j$, such that $s \in \mathscr{E}_j$. To define an MDP under this aggregated state space let the transition probabilities be (for meta-states $\mathscr{E}, \mathscr{E}'$ and action $a \in \mathcal{A}$):

$$\mathsf{P}(\mathscr{E}'|\mathscr{E}, a) = \frac{\mathsf{P}(s' \in \mathscr{E}', s \in \mathscr{E}, a)}{\mathsf{P}(s \in \mathscr{E}, a)}, \tag{4}$$

$$R(\mathscr{E}, a, \mathscr{E}') = \mathsf{E}(R(s, a, s')|s' \in \mathscr{E}', s \in \mathscr{E}, a). \tag{5}$$

We summarize the dynamics implied by Equations (4) and (5) with the notation $\mathsf{P}(\mathscr{E}', r|\mathscr{E}, a)$. A solution method of the aggregated MDP using Linear Programming is given by solving the following optimization program:

$$\min_{V,q} \sum_{j=1}^J \mu(\mathscr{E}_j) V(\mathscr{E}_j) \tag{6}$$

$$s.t. \quad q(\mathscr{E}_j, a) = \sum_{k=1}^J \sum_r \mathsf{P}(\mathscr{E}_k, r|\mathscr{E}_j, a)[r + \gamma V(\mathscr{E}_k)], \quad \text{for } j = 1, \ldots, J, \, a \in \mathcal{A},$$

$$V(\mathscr{E}_j) \geq q(\mathscr{E}_j, a), \quad \text{for } j = 1, \ldots, J, \, a \in \mathcal{A}.$$

Where $\mu$ is an arbitrary probability measure on the meta-states $\{\mathscr{E}_j\}$ such that $\mu(\mathscr{E}_j) > 0$ for all $j = 1, \ldots, J$ (see Chapter 6 of Puterman (1994)). Note that when the meta-states $\mathscr{E}$ are singletons, that is, each event only contains one state, this formulation is equivalent to solving the original MDP.

The policy implied by the solution of this MDP (denoted $\tilde{\pi}$) is naturally defined in the aggregated state space $\{\mathscr{E}_j\}_{j=1}^J$. That is, $\tilde{\pi}^* : \{\mathscr{E}_j\}_{j=1}^J \to \mathcal{A}$. Extending this policy to the original state space is straight-forward by letting $\tilde{\pi}^*(s) = \tilde{\pi}^*(\mathscr{E}_j)$ for all $s \in \mathscr{E}_j$. Whenever a policy is defined, its state space can be interchangeably understood by this extension.

## 3.4 Tree Partition and Approximation to Original MDP

A tree $\mathcal{T}$ is a collection of sequential binary partitions of the state space $\mathcal{S}$. For example, suppose $\mathcal{S} = \{x : x \in \mathbb{R}^d\}$, that is, the state space is the set of $d$-dimensional real-valued vectors, where $x_i$ is the $i$-th entry of the vector $x$. A tree $\mathcal{T}$ is composed by a set of $J$ leaves $\mathcal{L} = \{\mathscr{E}_j\}_{j=1}^J$. A leaf is a sequence of binary splits of the state space, for example, the leaf $\mathscr{E} = x_{i_1} \leq \tau_1, \cdots, x_{i_k} > \tau_k$ is composed by a sequence of binary splits of the state space ($\leq$ or $>$) at some threshold level sequence $\tau_1, \ldots, \tau_k \in \mathbb{R}$, with $i_1, \cdots, i_k$ being a index sequence of integers in $\{1, \cdots, d\}$. The set of leaves is a disjoint partition of the state space, that is $\cup_{j=1}^J \mathscr{E}_j = \mathcal{S}$ and $\mathscr{E}_j \cap \mathscr{E}_k = \varnothing$ for $j \neq k$.

Given a tree $\mathcal{T}$ with a set of leaves $\mathcal{L} = \{\mathscr{E}_j\}_{j=1}^J$ the problem in Equation 6 can be solved to obtain a solution of the aggregated MDP. Let $G(\pi)$ be the cumulative expected reward of the original MDP under policy $\pi$ and pdf $\lambda$ for the initial state (that is, $\lambda(s) \geq 0$ and $\sum_{s \in \mathcal{S}} \lambda(s) = 1$ with cdf $\Lambda(s)$ defined by some ordering of $s \in \mathcal{S}$), that is, let $G(\pi) := \mathsf{E}_\pi \left[ \sum_{t=0}^\infty \gamma^t R(s_t, a_t, s_{t+1}) | s_0 = \Lambda^{-1}(U) \right]$ where $U$ is an independent uniform $(0, 1)$ random variable.

An approximation of the expected reward $G$ given by the aggregated MDP can be obtained by setting the measure $\mu(\mathscr{E}_j)$ equal to $\mu(\mathscr{E}_j) = \sum_{s \in \mathscr{E}_j} \lambda(s)$. The intuition of why this measure approximates $G(\tilde{\pi})$ is given by the fact that the optimal objective value of the linear program $\sum_{j=1}^J \mu(\mathscr{E}_j) V_*(\mathscr{E}_j)$ is the expected reward of the aggregated MDP with starting state given by the distribution $\mu$. By making $\mu(\mathscr{E}_j)$ proportional to $\mathsf{P}(s_0 \in \mathscr{E}_j)$ in the original MDP we match the distribution of the initial state. As the dynamics $\mathsf{P}(\mathscr{E}', r | \mathscr{E}, a)$ are an average of the dynamics of the original MDP, we have $\tilde{G}(\tilde{\pi}^*) := \sum_{s \in \mathscr{E}_j} \mathsf{P}(s_0 \in \mathscr{E}_j) V_*(\mathscr{E}_j)$ is an approximation of $G(\tilde{\pi}^*)$ where $\tilde{\pi}^*$ is the policy implied by solving the LP in Equation (6). In Section A.4 we discuss the optimality of the aggregated policy.

## 3.5 Tree Algorithm for MDP

In this subsection we assume the state space is the set of $d$-dimensional real-valued vectors, that is, $\mathcal{S} = \{x : x \in \mathbb{R}^d\}$, where $x_i$ is the $i$-th entry of the vector $x$. Let $|\mathscr{E}|$ be the Lebesgue measure of a subset $\mathscr{E} \subseteq \mathbb{R}^d$. The meta-states $\mathscr{E} = x_{i_1} \leq \tau_1, \cdots, x_{i_k} > \tau_k$ are composed by a sequence of binary splits of the state space ($\leq$ or $>$) at some threshold level sequence $\tau_1, \ldots, \tau_k \in \mathbb{R}$, with $i_1, \cdots, i_k$ being a index sequence of integers in $\{1, \cdots, d\}$.

Given a tree $\mathcal{L}$, let $\tilde{G}(\mathcal{L})$ be the expected reward of the MDP after solving the aggregated MDP with partition $\mathcal{L}$ in Equation (6), that is, $\tilde{G}(\mathcal{L}) := \tilde{G}(\tilde{\pi}^*)$ (as discussed, this in an approximation of the reward of the original MDP). Let the initial partition $\mathcal{T}_0$ with leaves set composed of one single meta-state $\mathscr{E}_0 = \mathcal{S}$. The set of leaves is $\mathcal{L}_0 = \{\mathscr{E}_0\}$. As an improvement step, we are interest in further partitioning the state space from $\mathcal{L}_0$ into a partition $\mathcal{L}_1$ such that $\tilde{G}(\mathcal{L}_0) < \tilde{G}(\mathcal{L}_1)$, meaning that the expected reward of the MDP (under the partition $\mathcal{L}_1$) is higher than under $\mathcal{L}_0$. Intuitively, adding information allows to make better decisions. As $\mathcal{L}_0 \subset \mathcal{L}_1 \subset \cdots \subset \mathcal{L}_k$, meaning that the sets of leaves is grown from the set of leaves from the previous iteration. Then, a sequence of increasing expected reward partitions would be found, that is, a sequence:

$$\tilde{G}(\mathcal{L}_0) \leq \tilde{G}(\mathcal{L}_1) \leq \cdots \leq \tilde{G}(\mathcal{L}_{k-1}) \leq \tilde{G}(\mathcal{L}_k). \tag{7}$$

An interpretation of (7) is that a sequence of increasing partitions $\mathcal{L}_0 \subset \cdots \subset \mathcal{L}_k \subseteq \mathcal{S}$ each providing more information (and a more accurate model approximation) such that at every iteration, as more information is available, better policies can be found.

Selecting a partition can be done as follows: Start with a partition $\mathcal{L}$. Given the set of leaves $\mathcal{L}$, let $L(x_i \leq \tau) := \{\mathscr{E} \in \mathcal{L} : |\mathscr{E}| \geq |\mathscr{E}, x_i \leq \tau|\}$, that is, the set of leaves where partitioning at $x_i \leq \tau$ would effectively partition it (thus reducing its uniform measure in the original state space). Consider the candidate partition $\mathcal{L}(x_i, \tau) := \mathcal{L} \setminus L(x_i \leq \tau) \cup \bigcup_{\mathscr{E} \in L(x_i \leq \tau)} \{\mathscr{E}, x_i \leq \tau\} \cup \{\mathscr{E}, x_i > \tau\}$,

that is, removing the events $\mathscr{E}$ from the leaf set by splitting them into two new leaves (events)

$\{\mathscr{E}, x_i \leq \tau\}$ and $\{\mathscr{E}, x_i > \tau\}$. Then, greedily select the partition that most increases the expected reward of the MDP. That is, by optimizing the following problem:

$$\max_{i,\tau} \tilde{G}(\mathcal{L}(x_i, \tau)). \tag{8}$$

Which can be seen as selecting the variable $i$ and the threshold $\tau$ that best partitions the state space $\mathcal{L}$ and generates highest expected reward, each of these operations needs to compute the input probabilities for the LP in Equations (4) and (5) and the LP in Equation (6). After finishing the pass, the new partition (set of new leaves) is equal to $\mathcal{L} \setminus L(x_{i^*} \leq \tau^*) \cup \bigcup_{\mathscr{E} \in L(x_{i^*} \leq \tau^*)} \{\mathscr{E}, x_{i^*} \leq \tau^*\} \cup \{\mathscr{E}, x_{i^*} > \tau^*\}$

for $i^*, \tau^* = \operatorname{argmax}_{i,\tau} \tilde{G}(\mathcal{L}(x_i, \tau))$. The algorithm can be summarized as:

**Algorithm 1: State-Space Partition Algorithm for MDP**:

- Step 0: Initialize $\mathcal{L} = \{\mathscr{E}_0\}$. With $\mathscr{E}_0 = \mathcal{S}$. Let $g_* := \tilde{G}(\mathcal{L})$. Go to Step 1.

- Step 1: Solve $\max_{i,\tau} \tilde{G}(\mathcal{L}(x_i, \tau))$. Go to Step 2.

- Step 2: If $\max_{i,\tau} \tilde{G}(\mathcal{L}(x_i, \tau)) > g_*$ update $g_* \leftarrow \max_{i,\tau} \tilde{G}(\mathcal{L}(x_i, \tau))$, $\mathcal{L} \leftarrow (\mathcal{L} \setminus L(x_{i^*} \leq \tau)) \cup \bigcup_{\mathscr{E} \in L(x_{i^*} \leq \tau^*)} \{\mathscr{E}, x_{i^*} \leq \tau^*\} \cup \{\mathscr{E}, x_{i^*} > \tau^*\}$ for $i^*, \tau^* = \operatorname{argmax}_{i,\tau} \tilde{G}(\mathcal{L}(x_i, \tau))$ and go to Step 1. Otherwise, stop.

Each new partition may split more than one leaf node at every iteration. See the example in Section A.1 for a step by step example run of the algorithm in a 2-dimensional grid.

## 3.6 COMPUTATIONAL COMPLEXITY

As discussed in subsection 3.2 the optimal policy $\pi^*$ of the original MDP can be represented by a tree with $J^*$ leaves (approximating the optimal policy as $\pi^*(s) = \sum_{j=1}^{J^*} \tilde{\pi}(\mathscr{E}_j) \mathbf{1}\{s \in \mathscr{E}_j\}$). Suppose our procedure can find an equivalent tree after a certain number of iterations such that the size of the tree is equal to $J^*$ at such iteration $k$ (this depends on the particular instance and its transition and reward structure). In this section we discuss the computational effort at iteration $k$ of our algorithm and use this result to bound the overall computational complexity of our algorithm.

Solving an MDP using Linear Programming as in Equation (6) has complexity of order $O(n^p)$ where $n$ is the number of variables/constraints (in our LP formulation the number of constraints is equal to the number of variables in Equation (6) which is the size of the state-action space $|\mathcal{S}||\mathcal{A}|$). The degree of the polynomial can be taken as $p = 2.5$ if following the algorithm for LP in Vaidya (1989) or even lower when considering more recent advances in matrix multiplication as in Cohen et al. (2021).

The complexity of our procedure at iteration $k$ is the number of partitions that can be done at iteration $k$ times the complexity of solving instances of LP of size $n = s(k)|\mathcal{A}|$ (the size of the instances increases as the iterations of our algorithm increase). As the number of total possible partitions is bounded by some constant (dependent on the size of the original state space) $K(\mathcal{S})$, the complexity of our algorithm at iteration $k$ is $O((K(\mathcal{S}) - k)s(k)^p|\mathcal{A}|^p)$. That is, the number of available partitions $K(\mathcal{S}) - k$ times the complexity of solving a LP of size $s(k)$. As $s(k)$ is a worst-case estimate of the size of the state-action space, letting $k = s^{-1}(J^*)$ is a conservative estimate of the number of iterations necessary for the algorithm to find the tree.

The complexity can be then bounded as $s^{-1}(J^*)O((K(\mathcal{S}) - s^{-1}(J^*))s(s^{-1}(J^*))^p|\mathcal{A}|^p)$ which can be simplified to $O(K(\mathcal{S})J^{*p}|\mathcal{A}|^p)$ by absorbing $s^{-1}(J^*)$ as a constant inside the big-$O$ notation (noting that it is independent of the size of the state-action space). Comparing this with the complexity of the original MDP of order $O((|\mathcal{S}||\mathcal{A}|)^p)$ highlights the potential computational savings of our algorithm, as $K(\mathcal{S}) \ll |\mathcal{S}|$ (for intuition, consider the case when $\mathcal{S}$ is a discretized $d$-dimensional grid of size $|\mathcal{S}| = M^d$, across every dimension there are $M$ possible levels to partition, then the total number of partitions is $K(\mathcal{S}) = Md$).

Up to a constant, our procedure is more efficient than solving the original problem if $J^* < |\mathcal{S}|/\sqrt{K(\mathcal{S})}$. Or, in other words, up to a constant our procedure is $|\mathcal{S}|/\sqrt{K(\mathcal{S})}J^*$ faster than solving the original MDP. This highlights that the savings of our procedure happen obviously when

$J^*$ is small compared to $|\mathcal{S}|$. Note that the action space size $|\mathcal{A}|$ is always a bottleneck as the LP instances are of size always proportional to $|\mathcal{A}|$ at every iteration. Thus, our procedure is tractable as long as the action space $\mathcal{A}$ is relatively small. Likewise, the savings in computation also depend on the size $J^*$ of the tree which a priori is not known.

In the next two subsections we present an explicit from for the state space function size $s(k)$ and the corresponding complexity to the common "box" discretization scheme in 2 and $d$ dimensions.

### 3.6.1 2-DIMENSIONAL GRID

We describe the worst-case complexity of our algorithm a 2-dimensional grid: Let the state space $\mathcal{S} = \{(x,y) : x = 1, \ldots, M, y = 1, \ldots, N\}$ be a 2-dimensional grid of sizes $N$ times $M$. The size of the worst-case instances of the MDP at iteration $k$ is given by:

$$s(k) = \begin{cases} \left(\frac{k+2}{2}\right)^2 & k \leq 2m \\ k(m+1) - m^2 + 1 & k > 2m \end{cases} \quad (9)$$

for $m := N \wedge M$. The expression comes from dividing a rectangle using $k$ lines (partitions), $k_x$ vertical and $k_y$ horizontal, such that $k = k_x + k_y$ results in $(k_x + 1)(k_y + 1)$ partitions of a rectangle. This product is maximal when $q_x = q_y$ getting the bound on the number of partitions $\left(\frac{k}{2} + 1\right)^2 = \left(\frac{k+2}{2}\right)^2$. Next, as $k_x \leq N$ and $k_y \leq M$ without loss of generality assume $N < M$. When the number of partitions $k$ is greater than $2N$ we have $k_x$ remains at $N$ for a total of $(N+1)(k-N+1) = k(N+1) - N^2 + 1$ partitions.

In this case the number of new partitions available at every iteration is $K(\mathcal{S}) - k = N + M - k + 1$ (the minus $k$ comes from the fact that at every iteration there is one partition less available out of $N + M$ possible partitions). Then, for $k > 2m$ and without loss of generality $N \leq M$ such that $m = N$, the worst case complexity is $O((N + M - k + 1)(k(m+1)|\mathcal{A}|)^p)$ which can be simplified to $O(M(k(N+1)|\mathcal{A}|)^p)$ after canceling the $N + 1$ outside with $k$ that is greater than $2N$. Compare this to the complexity of the original problem of $O((NM|\mathcal{A}|)^p)$. In this case, our algorithm works best when $N$ is far from $M$ from a complexity point of view as we can take $M$ (which in this case is the maximum between $N$ and $M$) from inside the polynomial of the LP complexity and deal with a smaller size $2N < k < M$ instances of LP.

In summary, by letting $m = N \wedge M$ and $\bar{m} = N \vee M$ the worst-case complexity of our procedure is $O(\bar{m}(k(m+1)|\mathcal{A}|)^p)$ compared to original problem of complexity $O((NM|\mathcal{A}|)^p)$. See Section A.2 for the complexity analysis of the $d$-dimensional case. The $d$-dimensional grid case is also promising for applications where the state space is continuous (and thus infinite and uncountable) but some suggested discretization is possible (either data-driven or ad-hoc). In this case, while the state space is infinite, the partition algorithm can guide an optimal discretization structure. See the cartpole balancing example in the numerical section.

## 4 NUMERICAL EXAMPLES

### 4.1 GRIDWORLD

We present an instance of Gridworld as an example of our method. Gridworld consist of a grid where an agent moves to reach a goal (ending) state using one of four moves (actions) $\mathcal{A} = \{\uparrow, \downarrow, \rightarrow, \leftarrow\}$ (moving up, down, right or left one step in the grid). Each state is represented by a coordinate $(x, y) \in \mathbb{Z}^{2+}$ in a grid. Each state $(x, y)$ has a reward $r$ depending only in the state. In this setting transitions are deterministic once an action is taken (this can be relaxed without loss of generality, to allow jumps or stochastic transitions once an action is taken). We illustrate an instance of Gridworld in Figure 3.

In Figure 4 we plot side by side the optimal policy by solving the original MDP and our tree algorithm in **Algorithm 1**. Although the instance is trivial it highlights the computational savings of our methodology by only needing 6 states (other than the initial and terminal states which are given) rather than 28 in the original state space.

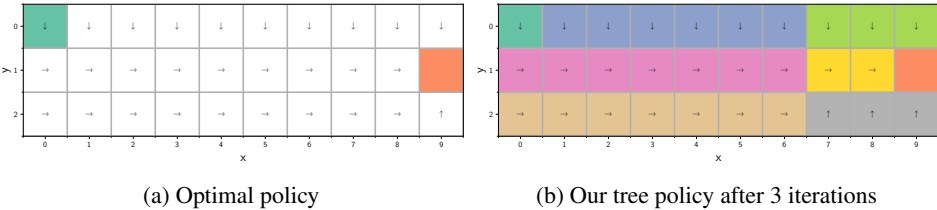

Figure 3: Gridworld example: The green square is the starting state and the orange state is the goal. Each number in the grid represents the reward of visiting that state. The goal (starting at any point) is to reach the terminal state incurring minimum cost.

| (a) Optimal policy | (b) Our tree policy after 3 iterations |
|---|---|

Figure 4: Side by side comparison of optimal policy $\pi^*$ and the tree policy implied by our algorithm.

## 4.2 CARTPOLE BALANCING

A popular benchmark for Reinforcement Learning problems is the so-called cartpole balancing problem, which consists of controlling a cart that can move either to right or left in a straight rail. Perpendicular to the axis of movement of the cart, there is a pole attached to a rotating axis at its center. With two actions $\mathcal{A} = \{\leftarrow, \rightarrow\}$ the goal is to have the pole attached to the cart balanced. The state space of the system is a 4-dimensional vector of real numbers $\mathcal{S} = \{(x, v, \theta, w) : x \in [-\ell_1, \ell_1], v \in [-\ell_2, \ell_2], \theta \in [-\ell_3, \ell_3], w \in [-\ell_4, \ell_4]\}$. In Figure 5 we present a diagram of the cartpole system. $x$ in the diagram represents the horizontal position of the pole, $v$ represents the horizontal velocity of the cart (positive means moving to the right), $\theta$ is the angle (in radians) from the vertical upright position of the pole (positive means the pole is in a clockwise position) and $w$ is the angular velocity of the pole (positive means rotating clockwise).

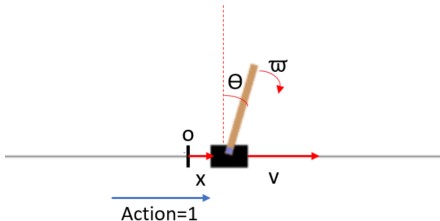

Figure 5: Cartpole state space and diagram.

In this case, we generate $n = 10,000$ random instances of data $\{(x, v, \theta, w), a, r, (x', v', \theta', w')\}$ with a totally random policy (given a state of the cart $(x, v, \theta, w)$, move the cart left or right with equal probability). The reward $r$ is equal to 1 for every unit of time the pole stays upright (the angle is $|\theta| < 0.28$ radians), otherwise the reward $r$ is 0 (meaning that the pole has fallen). Note that is very different from the usual pipeline of doing reinforcement learning as we only used a previous generated data and do not assume access to a simulator to generate new policies in this case. With this sample data, we estimate sample estimates of the parameters in Equations (4) and (5).

As it is, the problem is hard as the control problem is continuous on the state variables. Moreover, the equations governing the mechanics of the cart are non-linear equations that in principle are not given and subject to random perturbations. Typically, one way the infinite state space problem can be remedied is by discretizing it resulting in the $d$-dimensional grid case discussed in Section A.2

(with $d = 4$) with computational complexity $O(|M_1 M_2 M_3 M_4 2|^p)$ for the original MDP resulting in a large state space size even for moderate values of $M_1, M_2, M_3, M_4$.

Using our decision tree procedure in **Algorithm 1** for $M_i = 10$ equidistant cuts of the domain $[-\ell_i, \ell_i]$ for $i = 1, \ldots, 4$ solves the problem in 3 iterations of the algorithm (in this case, solving the cartpole problem consist in finding a policy that maintains the pole stable in the upright position beyond a threshold time). The size of the instances at the 3rd iteration of our algorithm is just 6 states for a total complexity of order $40(6 \times 2)^p$ using **Algorithm 1** compared to solving a 4-dimensional discretized MDP of complexity of order $(10^4 \times 2)^p$.

The policy that solves the cartpole problem is plotted in Figure 2. For $n' = 1000$ random unseen instances controlled with the policy in Figure 2 (started in the upright position with a random perturbation), the average expected reward is 187.93 with median 199.0 (the instances were stopped once the reward reached 200). It is important to note that this control tilts the cart slightly to the left, nonetheless, given the succinct policy representation it is noteworthy that the cartpole can be successfully controlled at all with a data-driven algorithm not deep learning based.

This method opens up a new angle to tackle non-linear continuous control problems without a simulator in the presence of uncertainty where the optimal policy can be represented as a lower dimensional tree policy. Moreover, the tree policy is attractive compared to a black-box policy. Doing an adversarial analysis (an adversarial attack consist in finding inputs in the state space that generate an undesirable action, see Behzadan & Munir (2017)) on a finite tree is much easier than with a continuous deep neural network. For example, a tree based policy used for landing a drone relying on sensors can be easily stress tested for reliability as the policy of the tree has a finite number of scenarios where an action is deterministically chosen. This cannot be said from a policy encoded in a black-box deep neural network as there can always be regions in the input space where the action chosen is unknown and an attacker can try to exploit the policy at these regions to induce catastrophic outcomes.

## 5    CONCLUSION

In this paper we have presented a decision tree algorithm for MDP that is computationally tractable even when the state space is prohibitively large as long as the optimal policy has a lower dimensional tree structure. On top of computational tractability the tree structure of the solution policy is transparent and interpretable.

Further research effort is needed in exploring the statistical properties of sampling partitions to speed up the algorithm (as in traditional tree algorithms). Another interesting open research question is to establish a theory for adversarial RL for tree-based policies. Moreover, there is an open path for extending these ideas to an online RL algorithm where an interpretable policy is desirable but exact methods are intractable as in the cartpole example.

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

## A    APPENDIX

### A.1    2-D GRIDWORLD EXAMPLE

We illustrate our algorithm with the following example: Let $\mathcal{S} = \{(x, y) : x = 1, \ldots, M, y = 1, \ldots, N\}$ be a 2-dimensional grid. In this 2-dimensional grid the action set is composed of four actions $\mathcal{A} = \{\uparrow, \downarrow, \leftarrow, \rightarrow\}$, that is, moving up, down, left or right in the grid. Moreover, let the rewards $R(x, y)$ arbitrary real values assigned to each point in the grid. The reward of the MDP is discounted at a rate $\gamma \in (0, 1)$. Let $\mathsf{P}((x', y'), r|(x, y), a)$ be the transition probabilities and rewards of the original problem (for example, if we consider the deterministic case where every action takes the agent to the intended square without perturbation, then, $\mathsf{P}((x, y + 1)|(x, y), \uparrow) = 1$ for $x = 1, \ldots, M, y = 1, \ldots, M - 1$).

At iteration 0 (initialization) we have $\mathscr{E}_0 = \mathcal{S}$, that is, all states are aggregated into a single state. Solving this "degenerate" MDP amounts to evaluate the following program:

$$
\begin{aligned}
\min_{V, q} \ & \mu(\mathscr{E}_0)V(\mathscr{E}_0) \\
s.t. \quad & q(\mathscr{E}_0, a) = R(\mathscr{E}_0, a, \mathscr{E}_0) + \gamma V(\mathscr{E}_0), \quad \text{for } a \in \mathcal{A}, \\
& V(\mathscr{E}_0) \geq q(\mathscr{E}_0, a), \qquad\qquad\quad \text{for } a \in \mathcal{A}.
\end{aligned}
$$

$R(\mathscr{E}_0, a, \mathscr{E}_0) = \mathsf{E}\{R(x', y') | (x', y') \in \mathscr{E}_0, (x, y) \in \mathscr{E}_0, a\}$ is the average reward of performing action $a \in \mathcal{A}$ in the aggregated state as in Equation (5). For example, $R(\mathscr{E}_0, \rightarrow, \mathscr{E}_0) = \frac{1}{(M-1)N} \sum_{i=2}^{M} \sum_{j=1}^{N} R(i, j)$, $R(\mathscr{E}_0, \leftarrow, \mathscr{E}_0) = \frac{1}{(M-1)N} \sum_{i=1}^{M-1} \sum_{j=1}^{N} R(i, j)$, $R(\mathscr{E}_0, \uparrow, \mathscr{E}_0) = \frac{1}{M(N-1)} \sum_{i=1}^{M} \sum_{j=2}^{N} R(i, j)$ and $R(\mathscr{E}_0, \downarrow, \mathscr{E}_0) = \frac{1}{M(N-1)} \sum_{i=1}^{M} \sum_{j=1}^{N-1} R(i, j)$, note how on each expected value, the boundary (of states that are not visited) is removed. The transition probabilities are all equal to one, as any action is going to move to the only state $\mathscr{E}_0 = \mathcal{S}$.

The above program can be further simplified as $V(\mathscr{E}_0) = (1 - \gamma)^{-1} \max_{a \in \mathcal{A}} R(\mathscr{E}_0, a, \mathscr{E}_0)$ as $\mu(\mathscr{E}_0) = 1$. This quantity is a rough approximation of the reward of the MDP by only performing the action $\arg \max_{a \in \mathcal{A}} R(\mathscr{E}_0, a, \mathscr{E}_0)$ at every state in the original state space $\mathcal{S}$.

At the next iteration, the goal is to split state $\mathscr{E}_0$ into two states $\mathscr{E}_1$ and $\mathscr{E}_2$ that has the highest increase in the expected reward of the MDP. The possible vertical splits are $\tau_x = 1, \ldots, M$ while the horizontal splits are $\tau_y = 1, \ldots, N$. Performing a split at level $\tau_x$ results in two states: $\mathscr{E}_1 = x \leq \tau_x$ and $\mathscr{E}_2 = x > \tau_x$. This results in the following set of equations:

$$R(\mathscr{E}_1, \uparrow, \mathscr{E}_1) = \frac{1}{\tau_x(N-1)} \sum_{i=1}^{\tau_x} \sum_{j=2}^{N} R(i, j), \quad R(\mathscr{E}_1, \downarrow, \mathscr{E}_1) = \frac{1}{\tau_x(N-1)} \sum_{i=1}^{\tau_x} \sum_{j=1}^{N-1} R(i, j),$$

$$R(\mathscr{E}_1, \rightarrow, \mathscr{E}_1) = \frac{1}{(\tau_x - 1)N} \sum_{i=2}^{\tau_x} \sum_{j=1}^{N} R(i, j), \quad R(\mathscr{E}_1, \leftarrow, \mathscr{E}_1) = \frac{1}{(\tau_x - 1)N} \sum_{i=1}^{\tau_x - 1} \sum_{j=1}^{N-1} R(i, j),$$

$$R(\mathscr{E}_2, \uparrow, \mathscr{E}_2) = \frac{1}{(M - \tau_x)(N-1)} \sum_{i=\tau_x+1}^{M} \sum_{j=2}^{N} R(i, j), \quad R(\mathscr{E}_2, \downarrow, \mathscr{E}_2) = \frac{1}{(M-\tau_x)(N-1)} \sum_{i=\tau_x+1}^{M} \sum_{j=1}^{N-1} R(i, j),$$

$$R(\mathscr{E}_2, \rightarrow, \mathscr{E}_2) = \frac{1}{(M - \tau_x - 1)N} \sum_{i=\tau_x+2}^{M} \sum_{j=1}^{N} R(i, j), \quad R(\mathscr{E}_2, \leftarrow, \mathscr{E}_2) = \frac{1}{(M-\tau_x-1)N} \sum_{i=\tau_x+1}^{M-1} \sum_{j=1}^{N} R(i, j),$$

$$R(\mathscr{E}_1, \rightarrow, \mathscr{E}_2) = \frac{1}{N} \sum_{j=1}^{N} R(\tau_x + 1, j), \quad R(\mathscr{E}_2, \leftarrow, \mathscr{E}_1) = \frac{1}{N} \sum_{j=1}^{N} R(\tau_x, j).$$

With transition probabilities:

$$\mathsf{P}(\mathscr{E}_1 | \mathscr{E}_1, \rightarrow) = \frac{(\tau_x - 1)N}{\tau_x N} = 1 - \tau_x^{-1}, \qquad \mathsf{P}(\mathscr{E}_2 | \mathscr{E}_1, \rightarrow) = \frac{N}{\tau_x N} = \tau_x^{-1},$$

$$\mathsf{P}(\mathscr{E}_2 | \mathscr{E}_2, \leftarrow) = \frac{(M - \tau_x - 1)N}{(M - \tau_x)N} = 1 - (M - \tau_x)^{-1}, \quad \mathsf{P}(\mathscr{E}_1 | \mathscr{E}_2, \leftarrow) = \frac{N}{(M - \tau_x)N} = (M - \tau_x)^{-1},$$

$$\mathsf{P}(\mathscr{E}_1 | \mathscr{E}_1, \uparrow) = \mathsf{P}(\mathscr{E}_1 | \mathscr{E}_1, \downarrow) = 1, \qquad \mathsf{P}(\mathscr{E}_2 | \mathscr{E}_2, \uparrow) = \mathsf{P}(\mathscr{E}_2 | \mathscr{E}_2, \downarrow) = 1.$$

With these, at the first iteration the LP in Equation (6) with two states is solved for $\tau_x$ (here we let $\mu(\mathscr{E}_1)$ be the uniform distribution, or simply said, the number of points in the grid in $\mathscr{E}_1$, that is $\tau_x N / MN$):

$$\min_{V, q} \frac{\tau_x N}{MN} V(\mathscr{E}_1) + \frac{(M - \tau_x)N}{MN} V(\mathscr{E}_2)$$

$$s.t. \quad q(\mathscr{E}_j, a) = \sum_{k=1}^{2} \mathsf{P}(\mathscr{E}_k | \mathscr{E}_j, a)[R(\mathscr{E}_j, a, \mathscr{E}_k) + \gamma V(\mathscr{E}_k)], \quad \text{for } j = 1, 2, \ a \in \mathcal{A},$$

$$V(\mathscr{E}_j) \geq q(\mathscr{E}_j, a), \qquad \qquad \text{for } j = 1, 2, \ a \in \mathcal{A}.$$

Solving an instance of this problem for each value of $\tau_x$ and $\tau_y$, and choosing the one with highest value for the LP is the first iteration of the algorithm. The objective is the approximated[2] expected reward of the original MDP by sampling the first state according to the distribution $\mu(\mathscr{E})$.

Suppose that the partition with highest expected reward (value of the LP) is given by $\tau_x^*$. The policy implied by the LP is $\pi^*(\mathscr{E}_1) = \arg \max_{a \in \mathcal{A}} q^*(\mathscr{E}_1, a)$ and $\pi^*(\mathscr{E}_2) = \arg \max_{a \in \mathcal{A}} q^*(\mathscr{E}_2, a)$.

---

[2]Under the aggregated space.

As discussed previously, the objective of the LP can be also seen as the approximated reward of the original MDP by following policy $\pi^*(\mathscr{E}_1)$ whenever $(x,y) \in \mathscr{E}_1$ and policy $\pi^*(\mathscr{E}_2)$ whenever $(x,y) \in \mathscr{E}_2$.

At next iteration a similar process is followed, given the states $\mathscr{E}_1 = x \leq \tau_x^*$ and $\mathscr{E}_2 = x > \tau_x^*$ at the end of the previous iteration, these states would be further partitioned by considering the rest of partitions $\tau_x = 1, \ldots, \tau_x^* - 1, \tau_x^* + 1, \ldots, M$ and $\tau_y = 1, \ldots, N$. Vertical partitions (across the $x$-axis) would partition the state space in 3 partitiosns (states) $\mathscr{E}_1 = x \leq \tau_x^*, x \leq \tau_x, \mathscr{E}_2 = x \leq \tau_x^*, x > \tau_x$ and $\mathscr{E}_3 = x > \tau_x^*$ for $\tau_x < \tau_x^*$ or $\mathscr{E}_1 = x \leq \tau_x^*, \mathscr{E}_2 = x > \tau_x^*, x \leq \tau_x$ and $\mathscr{E}_3 = x > \tau_x^*, x > \tau_x$ for $\tau_x > \tau_x^*$. Horizontal partitions (across the $y$-axis) would partition the state space into 4 partitions (states) given by: $\mathscr{E}_1 = x \leq \tau_x^*, y \leq \tau_y, \mathscr{E}_2 = x > \tau_x^*, y \leq \tau_y, \mathscr{E}_3 = x \leq \tau_x^*, y > \tau_y$ and $\mathscr{E}_4 = x > \tau_x^*, y > \tau_x$.

These two situations can be visualized by partitioning a rectangle vertically (across the $x$-axis) at first iteration, at the next iteration another vertical partition would result in 3 states, while a horizontal one (across the whole rectangle) would divide the state space into 4 regions. Whenever a partition is performed it might or not affect all states at the previous iteration, identifying which states are further divided is what motivates the definition of the set $L(x_i \leq \tau)$ in the algorithm subsection.

## A.2 $d$-DIMENSIONAL GRID

In the worst case of a $d$-dimensional grid, let the state space $\mathcal{S} = \{(x_1, \ldots, x_d) : x_1 = 1, \ldots, M_1, x_2 = \ldots, x_d = 1, \ldots, M_d\}$ be a grid of sizes $M_1 \times \cdots \times M_d$. Without loss of generality assume $M_1 \leq M_2 \leq \cdots \leq M_d$. Using a similar analysis as in the 2 dimensional case, a hyperrectangle in d-dimensions with $k = k_1 + \cdots + k_d$ lines partitioning it across every axes in every dimensions has $\prod_{j=1}^{d}(k_j + 1)$ partitions. Likewise, this product is maximal when $k_1 = \cdots = k_d$ resulting in at most $(k/d + 1)^d$ partitions at iteration $k$.

When $k > M_1/d$, we have that $k_1 = M_1$ and the number of partitions becomes $(M_1 + 1)\left(\frac{k - M_1}{d-1} + 1\right)^{d-1}$. By induction this process can be continued until the last dimension to get $\prod_{j=1}^{d-1}(M_j + 1)\left(k - \sum_{j=1}^{d-1} M_j + 1\right)$ partitions in the worst case. This result can be summarized with the function $s(k)$ as:

$$
s(k) = \begin{cases}
(k/d + 1)^d & k \leq M_1/d \\
(M_1 + 1)\left(\frac{k - M_1}{d-1} + 1\right)^{d-1} & M_1/d < k \leq M_2/(d-2) \\
(M_1 + 1)(M_2 + 1)\left(\frac{k - M_1 - M_2}{d-1} + 1\right)^{d-2} & M_2/(d-2) < k \leq M_3/(d-3) \\
\vdots & \\
\prod_{j=1}^{d-1}(M_j + 1)\left(k - \sum_{j=1}^{d-1} M_j + 1\right) & k > M_{d-1}/(d-1)
\end{cases} \tag{10}
$$

Similarly, the number of partitions available is $K(\mathcal{S}) - k = \left[\sum_{j=1}^{d} M_j\right] - k + 1$. Recall the state space is of size $|\mathcal{S}| = \prod_{j=1}^{d} M_j$. Depending on the structure of the problem, the worst possible case of the algorithm is of similar complexity to the original problem when $k$ is large. Nonetheless, in a usual instance of the problem the optimal partition selected by the algorithm does not follow the adversarial structure of these worst-case bounds of increasing the number of partitions as the objective value is rather trying to increase the overall reward of the MDP than just greedily increase the size of the state space.

## A.3 STATE AGGREGATION THEORY

We borrow the state aggregation framework of Li et al. (2006) to describe our proposed tree space aggregation procedure. A state aggregation of a given MDP $(\mathcal{S}, \mathcal{A}, P, R, \gamma)$ into an aggregated representation $(\tilde{\mathcal{S}}, \mathcal{A}, \tilde{P}, \tilde{R}, \gamma)$ is defined by a function $\phi$ mapping elements from the original MDP into new aggregated states, that is, $\phi : \mathcal{S} \to \tilde{\mathcal{S}}$. The transitions and rewards of the original MDP are linear combinations adding up to 1 of the original rewards, that is, for $\mathscr{E}, \mathscr{E}' \in \tilde{\mathcal{S}}$ (the elements of $\tilde{\mathcal{S}}$

are denoted as such as they can be interpreted as "events" of the original state space, for example, a leaf of a decision tree describes an event of the original state space and all states falling into it are aggregated as a single state):

$$P(\mathscr{E}'|\mathscr{E}, a) = \sum_{s \in \phi^{-1}(\mathscr{E})} \sum_{s' \in \phi^{-1}(\mathscr{E}')} w(s) P(s'|s, a), \tag{11}$$

$$R(\mathscr{E}, a, \mathscr{E}') = \sum_{s \in \phi^{-1}(\mathscr{E})} \sum_{s' \in \phi^{-1}(\mathscr{E}')} w(s) R(s, a, s'). \tag{12}$$

The weights $w(s)$ can be interpreted as the relative importance of each state into their aggregated state, these weights add to 1, that is $\sum_{s \in \phi^{-1}(\mathscr{E})} w(s) = 1$ for all $\mathscr{E} \in \tilde{\mathcal{S}}^3$. In Li et al. (2006) depending on properties of the aggregation function $\phi$ a hierarchy of state aggregations is presented. This hierarchy has on one side the coarsest possible aggregation where all states are collapsed to one, and on the other end the finest possible aggregation of the original state space (that is, the original representation of the state space). They describe a partial ordering between these aggregations (one aggregation contains another if all its partitions are subsets of the partitions of the other state aggregation, this is denoted by $\succeq$ where the finer aggregation scheme goes on the l.h.s. of the relationship). The hierarchy showed in the paper is summarized as (Theorem 2 in the paper):

$$\phi_0 \succeq \phi_{\text{model}} \succeq \phi_{Q^\pi} \succeq \phi_{Q^*} \succeq \phi_{a^*} \succeq \phi_{\pi^*}. \tag{13}$$

Where $\phi_0$ is the original (finest) representation of the MDP. $\phi_{\text{model}}$ is a representation where the rewards and transition probabilities of the aggregated states is the same as in the original MDP (the so-called bisimulation aggregation in the literature). $\phi_{Q^\pi}$ is an aggregation where all states aggregated have the same $Q$-function value for all policies $\pi$. In $\phi_{Q^*}$ all aggregated states have the same $Q$-function value for the optimal policy only and all actions $a \in \mathcal{A}$. $\phi_{a^*}$ is an aggregation where all aggregated states have the same optimal action and the optimal $Q$-function evaluated at the optimal action is the same for all states aggregated. Lastly, $\phi_{\pi^*}$ is an aggregation where all states in a class have the same optimal action.

This is an important discussion, because going to the right of the hierarchy (coarser aggregations of the state space) would achieve higher computational savings if solving an aggregated representation of the MDP allows to retrieve an equivalent solution to the original one. In Li et al. (2006) is showed that up to the aggregation scheme $\phi_{a^*}$ there are guarantees of optimality by solving the problem in the aggregated state space using $Q$-learning. There is also a negative result for the aggregation $\phi_{\pi^*}$ where convergence to the optimal policy of the original MDP is not guaranteed.

In this paper we propose another family of aggregations that we call $\phi_{tree}$, which is an abstraction where the original solution of the MDP has a tree structure and a reduced (aggregated) MDP where the states are determined by the leaf of such tree and its optimal solution is equivalent to the original MDP. That is, a MDP $(\mathcal{S}, \mathcal{A}, P, R, \gamma)$ has a solution $\pi^*(s) = \sum_{j=1}^{J} \mathbf{1}\{s \in \mathscr{E}_j\} \pi^*(\mathscr{E}_j)$ where $\pi^*(\mathscr{E}_j)$ for $j = 1, \ldots, J$ is the solution of the MDP $(\tilde{\mathcal{S}}, \mathcal{A}, \tilde{P}, \tilde{R}, \gamma)$ and $\{\mathscr{E}_j\}_{j=1}^{J}$ are the leaf of a tree composed of binary partitions.

The family of partitions $\phi_{tree}$ is of computational interest as it is between $\phi_{a^*}$ and $\phi_{\pi^*}$ in the partial ordering hierarchy of state aggregations, that is, $\phi_{a^*} \succeq \phi_{tree} \succeq \phi_{\pi^*}$, thus, potentially achieving sizable computational savings for large state MDPs beyond the well studied first 3 hierarchies ($\phi_{\text{model}}, \phi_{Q^\pi}$ and $\phi_{Q^*}$) trying to aggregate the state space according to the value function.

## A.4 Discussion on Optimality of Algorithm 1

Here we present an informal discussion of the convergence of **Algorithm 1** and present an intuitive argument of when the structure of the original MDP lends itself to have a tree structure such that the reduced MDP has an equivalent solution. In plain words, whenever the optimal structure (geometry of the paths) of the MDP is characterized by a (sub-)modular function, a tree structure of an aggregated MDP will capture the structure of the optimal solution of the original problem.

---

[3]These can be seen as analogous of Equations 4 and 12 for a policy $\pi_w$ with stationary distribution $w$ in that aggregation.

Consider the following gridworld example with state space $\mathcal{S} = \{(x, y) : x = 1, \ldots, M, y = 1, \ldots, N\}$ and reward equal to 1 at $(M, N)$. With action set $\mathcal{A} = \{\uparrow, \downarrow, \leftarrow, \rightarrow\}$. In this case the solution of this MDP is given by the value function $V_*(x, y) = \gamma^{d_1(x,y)}/(1 - \gamma^2)$ where $d_1(x, y)$ is the Manhattan distance to the point $(M, N)$ with reward equal to 1. To see why this is the case, the solution from any point $(x, y)$ is to take the deterministic shortest path to the point with reward 1, once this point is reached a reward of 1 is accumulated at even period, that is a reward $\gamma^{d_1(x,y)}(1 + \gamma^2 + \gamma^4 + \gamma^6 + \cdots)$ which is a geometric series by the substitution $\gamma' = \gamma^2$. What is important here is that the structure of the optimal policy with respect to the other states follows a structure determined by the distance function. To this end, define the random variable $T : \mathcal{S} \times \mathcal{A} \times U \to \mathcal{S}$ which denotes the transition to a new state by applying an action to the current state (the $U$ is a random uniform representing the randomness in the case where transitions are not deterministic). In this case $T_a(x, y)$ is just the deterministic state that is reached by applying action $a$ from state $(x, y)$, for example, action $a = \rightarrow$, makes $T_a(x, y) = (x + 1, y)$. With this, we have that the optimal policy is determined completely by the distance function $d_1(x, y)$ in the neighboring points spanned by the actions $a \in \mathcal{A}$:

$$\pi^*(x, y) = \arg\min_{a \in \mathcal{A}} 1 + d_1(T_a(x, y)), \tag{14}$$

This follows from $d_1(x, y) = 1 + d_1(T_{a^*}(x, y))$ with $a^* = \arg\min_{a \in \mathcal{A}} d_1(T_a(x, y))$ because the distance to any contiguous transition is equal to 1. The key observation is that while there are many shortest paths to the high reward circuit, the metric implied by $T_a(x, y)$ defines an aggregation of the policies as in Figure 6. This follows from the fact that the distance function is modular $d_1(x, y) = 1 + d_1(T_{a^*}(x, y))$ as previously discussed.

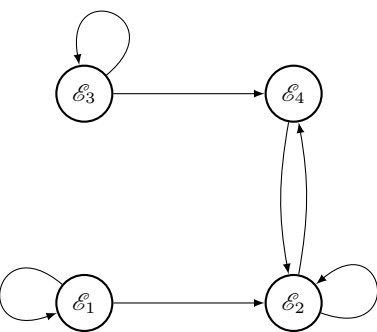

Figure 6: The point $(M, N)$ with reward 1 represented by a $\bullet$. A configuration of the shortest path trajectory such that points in every partition have the same direction. This is optimal as the metric induced by $d_1(x, y)$ is the Manhattan distance.

Then, an MDP aggregation with 4 states as in Figure 7 has the same structure in its solution than the original MDP. For any point $(x, y)$, the mapping to the aggregated states is given by $\mathscr{E}_1 = x < M, y < M$, $\mathscr{E}_2 = x \geq M, y < M$, $\mathscr{E}_3 = x < M, y \geq M$ and $\mathscr{E}_4 = x \geq M, y \geq M$, or simply the blue lines in Figure 6. As said before, Equations 4 and 5 can be seen as linear combinations (adding to 1) of the original rewards and transition probabilities. Then, the only state with positive reward is $\mathscr{E}_4$, that is, $R(\mathscr{E}_4) = 1$ for preceding any action. Otherwise $R(\mathscr{E}) = 0$ for $\mathscr{E} \in \{\mathscr{E}_1, \mathscr{E}_2, \mathscr{E}_3\}$.

Figure 7: Optimal policy $\pi^*$ in the aggregation of MDP in Figure 7.

The structure of the solution of this MDP is given by:

$$\pi^*(\mathscr{E}) = \arg\min_{a \in \mathcal{A}} \mathsf{E}[\tilde{d}(\mathscr{E}, T_a(\mathscr{E})) + \tilde{d}(T_a(\mathscr{E}), \mathscr{E}_4)] = \arg\min_{a \in \mathcal{A}} \mathsf{E}[p_a^{-1} + \tilde{d}(T_a(\mathscr{E}), \mathscr{E}_4)], \tag{15}$$

where $\tilde{d}(T_a(\mathscr{E}), \mathscr{E}_4)$ is the expected number of steps to state $\mathscr{E}_4$ from state $T_a(\mathscr{E})$. In this instance of the problem, for any two connected states $\mathscr{E}$ and $\mathscr{E}'$ by action $a \in \mathcal{A}$ (for example, from $\mathscr{E}_1$ to $\mathscr{E}_3$ only the action $a = \uparrow$ has positive probability of transitioning), with probability $p_a$ the chain moves to $p_a$, otherwise it stays with probability $1 - p_a$. From this, by the average of a geometric distribution the average number of steps to connected states is $\tilde{d}(\mathscr{E}, T_a(\mathscr{E})) = p_a^{-1}$. The structure of the aggregated MDP is optimal with respect to the original state space as $R(\mathscr{E}_4)$ is the only state with positive reward. Moreover, $\pi^*(x, y) = \pi^*(\mathscr{E})$ for $(x, y) \in \mathscr{E}$. Note that there is also a correspondence between the transient states of the optimal policy of the aggregated MDP and the transient states of the original MDP. The recurrent states of the aggregated MDP contain the recurrent of the original MDP.

This argument can be extended to problems where the optimal policy in both aggregated and original state spaces is given by two modular functions $d, \tilde{d}$ that captures the span of the actions and the recurrent circuits of maximum reward (given any policy $\pi$, the MDP behaves as a Markov Chain with transition matrix $P_\pi$. Then, the solution of the MDP is a Markov Chain that can be canonically decomposed into transient and closed recurrent classes, these classes are by definition high reward circuits). In a way, the modularity of the function $d$ is what allows the reduction of the state space to work.

As a last and key point, note that the optimal value function $V_*(x, y) = \gamma^{d_1(x,y)}/(1 - \gamma^2)$ is in a way a distraction from the true driver of the solution of the problem, the distance function $d_1(x, y)$. From this point of view, aggregating states according to the similarity of the value function is not the appropriate way of solving these instances.

