# OpenReview forum: "Decision Tree Algorithms for MDP"
_ICLR.cc/2022/Conference — ICLR 2022 Submitted_

### Official Review · Reviewer_qMvr · 2021-10-28

**Correctness:** 4
**Technical Novelty And Significance:** 3
**Empirical Novelty And Significance:** 3
**Recommendation:** 6
**Confidence:** 3

**Main Review:**

* The problem addressed by the paper - computationally tractable solutions to MDPs - is important. The submission is clear and, as far as I can see, correct.

* The interpretability of the resulting policy is a clear strength of the method.

* The rigorous analysis of the computational complexity is a strength of the paper.

* Empirical analysis: On the one hand, I find the analysis convincing in so far that the method needs very few (!) state partitions to solve these problems. On the other hand, there are no comparisons to baselines on these problems.

* The computational burden of repeatedly solving the Linear Programs with growing complexity is probably the main disadvantage and prevents you from affording many iterations. However, for MDPs where the optimal policy actually has a lower-dimensional tree structure, you won’t need that many iterations in turn. The paper clearly states that the method is dedicated to those problems only, so I think it is fine.

* Novelty: The authors mention that the method is similar to the Trie Model from Moore (1991), but I am unfamiliar with it and don't have access to the Moore (1991) paper, so I cannot assess this.

In general, the paper is well written. However, a few minor comments:
* Section 3.2, 2nd paragraph: „an analogous function approximation theorem for decision trees …“ I would appreciate a reference to this theorem in literature.
*Section1, 2nd paragraph: typo: the use OF exact methods, …
*Section 3.2, 4th paragraph: typo: …with respect to findinG optimal policies …
*Section 3.5, 1st paragraph: repetitive to Section 3.4 1st paragraph
*I would change the title. It says „Decision Tree AlgorithmS“, but there is only one algorithm in the paper. Maybe something like „A Decision Tree Algorithm for MDPs“ or „Decision Trees for MDPs“?

------
Post-rebuttal:

I understand why there are concerns about the scalability of the algorithm. In fact, I believe the method doesn’t scale for some problems. Namely all of the problems where the optimal policy doesn’t not have the underlying tree structure. But if the problems are of that kind, the method will only need few iterations and then it doesn’t matter too much if the iterations are expensive.

I agree that the experimental section lacks a comparison to baselines. And that the examples are simplistic. While I like the simplicity of the example problems (because they demonstrate the concept well), I can see that it is common for a conference like this to also include larger-scale experiments.

Reviewer vkFy pointed out that the benefit of having a decision tree structure for a policy is the interpretability and that in the grid and cart pole example there is not really a need for this. I agree, but another benefit of the decision tree structure is the potential decrease in number of training iterations and that’s what the gird and cart pole example demonstrate well.

I cannot comment on the novelty of the approach as I am not familiar with some parts of the related work. The other reviews might have a point here, or not.

I agree with the optimality issue described by reviewer HjwV, and also with the author’s answer to this issue. I don’t know if it is enough to include a section informally discussing it as proposed by the authors, or if it is necessary to fully formalize this insight.

In total, I think that the approach has potential and is worth to be published at some point, but addressing the raised points from the other reviewers could indeed further improve the paper.


**Summary Of The Paper:**

The authors propose a method to solve MDPs with large state-space and moderately sized action-space, where a decision tree can represent the optimal policy.
The decision tree is learned by iteratively partitioning the state space such that the approximation of the policy by the expanded decision tree improves the expected reward. Finding the best partition in each step involves solving a Linear Programm.
The authors provide an extensive analysis of the computational complexity of their method and demonstrate its strengths in a Gridworld example and the Cartpole problem.

**Summary Of The Review:**

I vote for acceptance as the described method is an intuitive approach to exploit lower dimensional structure to solve computationally difficult high-dimensional problems.

---

> ### Author Response · Authors · 2021-11-23
> **Reply to review.**
>
> Thank you for the much appreciated feedback on the manuscript.
>
> First of all, thank you for reading and evaluating the paper. We are glad to our message got across and that you share our enthusiasm for the idea.
>
>  >Section 3.2, 2nd paragraph: „an analogous function approximation theorem for decision trees …“ I would appreciate a reference to this theorem in literature.
>
> See the book "A Probabilistic Theory of Pattern Recognition" by Devroye, Luc, László Györfi, and Gábor Lugosi. Springer Vol 31. The results related to trees are in Ch 20.

---

### Official Review · Reviewer_HjwV · 2021-11-02

**Correctness:** 3
**Technical Novelty And Significance:** 2
**Empirical Novelty And Significance:** Not applicable
**Recommendation:** 3
**Confidence:** 2

**Main Review:**

The idea is simple and the computational complexity analysis, though not surprising, is convincing.
What is missing is an estimate of the efficacy of the decision tree approximation. In general greedy construction of decision trees is not optimal and it is a good approximation when some submodular property of the objective function can be assumed. The authors seem to completely ignore the issue of optimality. They talk about an approximation but, as far as I understand, unless I am missing something they do not consider whether the approximation is good or bad.


**Summary Of The Paper:**

The paper presents an approach to the MDP based on the construction of a decision tree that is expected to well approximate the optimal policy while reducing the complexity of computing such an effective policy.
The idea is to partition the state space into meta states and use a single action for all the states in the same class. The classical greedy construction of decision tree is applied to determine the partitioning. The splitting criterion is to choose at each iteration among all present leaves the one that allows the splitting with maximum gain in terms of expected reward.

**Summary Of The Review:**

I think that without some comment about the issue of how well the obtained decision tree based policy approximate the optimal LP based policy, the paper is weak. It present a nice but obvious explanation of why by coarsening the state space one can reduce the complexity of computing a good policy.

---

> ### Author Response · Authors · 2021-11-23
> **Reply to review.**
>
> Thank you for the much appreciated feedback on the manuscript.
>
> >What is missing is an estimate of the efficacy of the decision tree approximation. In general greedy construction of decision trees is not optimal and it is a good approximation when some submodular property of the objective function can be assumed. The authors seem to completely ignore the issue of optimality. They talk about an approximation but, as far as I understand, unless I am missing something they do not consider whether the approximation is good or bad.
>
> You are on point. We added a section discussing this [A.4] and you are exactly on the right track, a (sub-) modularity condition on the paths of the optimal policy are needed to ensure that the proposed tree aggregation would result in a equivalent optimal policy in the original state space. Formalizing these insights is indeed a worthwhile task, but we conjecture that the family of MDP instances that can be solved by reduced tree policies comprises many problems of interest. For example, when the actions are uniform and local across the state space.
>
> Short of a proof the matter is far from settled, but it is promising that for example the cartpole (that has infinite state space) can be controlled by a tree policy consisting only of 6 states. As a proof of concept, this might suggest that related robotics problems can be solved by compact tree controls found by similar heuristics as the one proposed.

---

### Official Review · Reviewer_vkFy · 2021-11-02

**Correctness:** 3
**Technical Novelty And Significance:** 2
**Empirical Novelty And Significance:** 2
**Recommendation:** 5
**Confidence:** 4

**Main Review:**

The insight that solving a partitioned MDP is an approximate solution to the full MDP is not particularly novel and is presented in [1] - which I think really should be referenced in the paper. The originality then rests on connecting decision trees to partitions of the state space (again something that has been noted before, but also the method for building up the partition which I do think has some originality in its focus.

This is a batch RL method when the state space becomes large (i.e. the grey region of Figure 1) as far as I can see, that is to say this is not a method that can be deployed online in the environment and learn as it goes, it needs to have seen data from somewhere else. This I don't think is made particularity clear in the paper and should be included in a discussion.

The experimental section of the paper is not at the level it needs to be, the method should be demonstrated on more than just a grid world environment and cart pole, both of which are extremely toy problems that are easily solved. Comparisons should also be made to existing works in the literature - of which there are obviously many.

Realistically the benefits of having a decision tree structure for a policy is so that the policy is interpretable and can be inspected and communicated well. There's not really any need for this in either a grid world or cart pole - a medical example on the other hand where you extracted a decision tree policy of a clinician (treating sepsis is a common example) would be really interesting.

Figure 1 is taking up far too much space for what it is demonstrating.

[1] Kim, Kee-Eung, and Thomas Dean. "Solving factored MDPs using non-homogeneous partitions." Artificial Intelligence 147.1-2 (2003): 225-251.

----- Post rebuttal -----

I thank the authors for their responses to my points, they make sense and I think including these explanations in the paper would aid in framing and justifying some of these decisions. Consequently I think slightly raising my score is fair, although I would still be hesitant to recommend acceptance without seeing a paper with a more thorough comparison to related work both textually and experimentally.

**Summary Of The Paper:**

The paper essentially draws on the insight that representing a policy as a decision tree is essentially the same as partitioning the state space and solving a simplified MDP on this less coarse space, and they present a method for then learning this partition before demonstrating results on a simple grid world example and the traditional cart pole control problem.

**Summary Of The Review:**

I do not think the paper is ready for publication in its present form although the method is potentially promising. An expanded discussion on related work is needed alongside a much more developed experimental section.

---

> ### Author Response · Authors · 2021-11-23
> **Reply to review.**
>
> Thank you for the much appreciated feedback on the manuscript.
>
> >The insight that solving a partitioned MDP is an approximate solution to the full MDP is not particularly novel and is presented in [1] - which I think really should be referenced in the paper. The originality then rests on connecting decision trees to partitions of the state space (again something that has been noted before, but also the method for building up the partition which I do think has some originality in its focus.
>
> First, thank you for pointing out this reference, it is indeed related to our procedure and we have added to our literature review. A key difference with this work is that our method is using an estimate of the overall performance of the MDP in the original state space. While the work mentioned is choosing partitions by diminishing the maximum discrepancy between value functions in between iterations. We believe this puts this work closer to the variable resolution literature. On this point we added a state aggregation theory section in the appendix to correctly position our proposed state aggregation methodology.
>
> >This is a batch RL method when the state space becomes large (i.e. the grey region of Figure 1) as far as I can see, that is to say this is not a method that can be deployed online in the environment and learn as it goes, it needs to have seen data from somewhere else. This I don't think is made particularity clear in the paper and should be included in a discussion.
>
> Yes, in principle the method presented works with estimates (from a batch) of the transition probabilities and rewards. Adapting the method to an online setting can be done by a meta algorithm generating new samples from the associated algorithm policy until convergence. Nonetheless, the efficiency and optimality of this procedure would require a deeper study and perhaps redesign of the current algorithm.
>
> >The experimental section of the paper is not at the level it needs to be, the method should be demonstrated on more than just a grid world environment and cart pole, both of which are extremely toy problems that are easily solved.
>
> The two numerical examples are presented for a reason: the first one on the gridworld for pedagogical purposes on visualizing what we mean by a tree partition and analyzing the algorithm. The second one as a example with infinite state space solved using only 6 states without having to approximate the $Q$-function at infinitely many points. While we agree that more computational examples would be welcome. As a proof of concept these two serve their purpose.
>
> >Realistically the benefits of having a decision tree structure for a policy is so that the policy is interpretable and can be inspected and communicated well. There's not really any need for this in either a grid world or cart pole
>
> As in the previous point, the cart-pole example highlights how an infinite dimensional MDP can be successfully controlled by a representation with only 6 states. In robotics having a finite and compact policy is of paramount importance for reliability. Especially is scenarios where a robotic system might interact with humans. Inspecting a black-box policy (such as a one derived from deep Q-learning) at infinitely many points is much more difficult than at 6 states (or an associated compact tree policy).
>
> > a medical example on the other hand where you extracted a decision tree policy of a clinician (treating sepsis is a common example) would be really interesting.
>
> Totally agree, we are working on a example of this kind in a journal version of this work.
>
> > Figure 1 is taking up far too much space for what it is demonstrating.
>
> While we agree on the size, the message of the image is an important part of the paper. That in the gray area there are families of instances of MDP where their structure can be exploited by exact methods and compact representations rather than asserting that deep Q-learning is a "one size fits all" method for RL.

---

### Official Review · Reviewer_MpCH · 2021-11-02

**Correctness:** 4
**Technical Novelty And Significance:** 1
**Empirical Novelty And Significance:** 1
**Recommendation:** 3
**Confidence:** 4

**Main Review:**


My first concern here is the novelty. As the paper misses a major related work: Tree-Based Batch Mode Reinforcement Learning, D. Ernst et. al., the tree architecture has been studied extensively before.
Secondly, the experiment section are not sufficient because a) they do not highlight if the algorithm is scalable since only grid world/cart-pole domains are presented b)there is no comparison with existing algorithms.
Thirdly, I do not agree with Fig. 1. Algorithms like DQN are in fact in the gray area where the state space is huge but the action space is just 4-dimensional. So this goes back to my previous points of scalability and comparison with existing algorithms like DQN.

I believe using trees in MDPs is good alternative to DNN, particularly because of their interpretability. It would be great if authors could comment/explore this. Furthermore, algorithms like monte-carlo trees provide a more adaptive search to zoom into "important" regions.

**Summary Of The Paper:**

This paper studies using decision tress in lieu of deep neural networks in MDPs. The authors use them to partition the state space and represent the policies. Then they use LP to find the optimal policy. The authors claim to tackle the "gray area" where the state space is huge but not large action spaces as one of their key contributions.



**Summary Of The Review:**

Trees are a great function approximators and have been studied in MDPs before.
Recent successful algorithms are based on deep neural nets but authors do not compare the two approaches in terms of their scalability, interpretability and/or theoretically.

---

> ### Author Response · Authors · 2021-11-23
> **Reply to review.**
>
> Thank you for the much appreciated feedback on the manuscript.
>
> >My first concern here is the novelty. As the paper misses a major related work: Tree-Based Batch Mode Reinforcement Learning, D. Ernst et. al., the tree architecture has been studied extensively before.
>
> We appreciate the reference as we missed it during the literature review. We agree that the work is related as it uses decision trees in a similar batch setting to solve MDPs, yet this falls into the value/$Q$ function approximation literature where samples are used to update the $Q$-function values at sample points $\{ s_i,a_i\ }$ and the tree function is used to interpolate/approximate the $Q$-function at unseen points. This is also the category that Deep Q-learning falls into.
>
> Our work follows a different stream of literature on directly aggregating the state space of the original MDP into a smaller representation, and solving the MDP in this smaller state space to generate a close to optimal policy in the original state space.
>
> >the experiment section are not sufficient because a) they do not highlight if the algorithm is scalable since only grid world/cart-pole domains are presented
>
> On the scalability note, we emphasize that the number of iterations/complexity depends on the structure of tree associated with the optimal policy (the number of leaves $J$ of the tree associated with the optimal policy). The complexity analysis section presents a picture of the scalability of the algorithm as a function of $J$ and the well-known complexity of solving LP instances.
> The two numerical examples are presented for a reason: the first one on the gridworld for pedagogical purposes on visualizing what we mean by a tree partition and analyzing the algorithm. The second one as a example with infinite state space $|\mathcal{S}|=\infty$ solved using only 6 states without having to approximate the $Q$-function at infinitely many points. While we agree that more computational examples would be welcome. As a proof of concept these two serve their purpose.
>
> >b)there is no comparison with existing algorithms
>
> This is fair in the sense that at the end of the day we are suggesting a policy generation procedure that can be bench-marked against other methods. We will incorporate these in a extended version of the manuscript.
>
> >Thirdly, I do not agree with Fig. 1. Algorithms like DQN are in fact in the gray area where the state space is huge but the action space is just 4-dimensional. So this goes back to my previous points of scalability and comparison with existing algorithms like DQN.
>
> This is in fact related to all the previous points. While we agree that DQN is the only tool available to tackle very high dimensional problems (both in state and action size), in the gray area there are still big families of MDP instances where there are more efficient algorithms, where rather than iteratively doing $Q$-learning exploiting the structure of the MDP allows to use exact methods. We added a Section on state aggregation theory in the appendix to illustrate the potential computational savings of MDP abstractions.
>
> >I believe using trees in MDPs is good alternative to DNN, particularly because of their interpretability. It would be great if authors could comment/explore this. Furthermore, algorithms like monte-carlo trees provide a more adaptive search to zoom into "important" regions.
>
> This is a very good point, we also believe that importance sampling related techniques can enhance the potential convergence speed of our procedure. Moreover, we also agree on the interpretability of trees as a key strength for many applications where a black-box policy is undesirable.

---

### Decision · Program_Chairs · 2022-01-20

**Decision:**

Reject

**Comment:**

The reviewers identified missing comparisons to existing baselines (Deep RL and other tree-based RL methods) as well as simplistic experiments as the main limitations of the paper. While the authors could address some of the issues raised by the reviewers, the missing comparisons and too simple experiments remain. I therefore agree with (most of) the reviewers that the paper can not be published at its current state.